:PLOS | ONE

# A global assessment of street-network sprawl

Christopher Barrington-Leigh[1]*, Adam Millard-Ball[2]

**1** Institute for Health and Social Policy and McGill School of Environment, McGill University, Montreal, Quebec, Canada, **2** Environmental Studies Department, University of California, Santa Cruz, California, United States of America

☼ These authors contributed equally to this work.
* Chris.Barrington-Leigh@McGill.ca

**Data Availability Statement:** The primary data underlying this study are third-party data available from OpenStreetMap at https://planet.openstreetmap.org. All other data used to aggregate the results are also third-party. These

## Abstract

Disconnected urban street networks, which we call "street-network sprawl," are strongly associated with increased vehicle travel, energy use and $CO_2$ emissions, as shown by previous research in Europe and North America. In this paper, we provide the first systematic and globally commensurable measures of street-network sprawl based on graph-theoretic and geographic concepts. Using data on all 46 million km of mapped streets worldwide, we compute these measures for the entire Earth at the highest possible resolution. We generate a summary scalar measure for street-network sprawl, the Street-Network Disconnectedness index (SNDi), as well as a data-driven multidimensional classification that identifies eight empirical street-network types that span the spectrum of connectivity, from gridiron to dendritic (tree-like) and circuitous networks. Our qualitative validation shows that both the scalar and multidimensional measures are meaningfully comparable within and across countries, and successfully capture varied dimensions of walkability and urban development. We further show that in select high-income countries, our measures explain cross-sectional variation in household transportation decisions, and a one standard-deviation increase in SNDi is associated with an extra 0.25 standard deviations in cars owned per household. We aggregate our measures to the scale of countries, cities, and smaller geographies and describe patterns in street-network sprawl around the world. Latin America, Japan, South Korea, much of Europe, and North Africa stand out for their low levels of street-network sprawl, while the highest levels are found in south-east Asia, the United States, and the British Isles. Our calculations provide the foundation for future work to understand urban processes, predict future pathways of transportation energy consumption and emissions, and identify effective policy responses.

## 1 Introduction

The global share of land under urban use is likely to nearly triple between 2000 and 2030 [1], as the human population becomes increasingly urbanized. The environmental impacts of this urban expansion, however, will largely be shaped by the finer-grained physical form of cities, not just the size and rates of growth of urban areas. While urban megaregions such as Mexico

additional data sources are the Database of Global Administrative Areas at https://gadm.org, the Global Human Settlement Layer at https://ghsl.jrc.ec.europa.eu/data.php, and the Atlas of Urban Expansion http://www.atlasofurbanexpansion.org. For more information about the specific data used in this study, please see the Supporting Information.

**Funding:** Our work was supported by Social Science and Humanities Research Council of Canada grant 435-2016-0531 (www.sshrc-crsh.gc.ca; CBL), the Hellman Fellows Program (www.hellmanfellows.org; AMB), and a UC Santa Cruz Faculty Research Grant (AMB). The funders had no role in study design, data collection and analysis, decision to publish, or preparation of the manuscript.

**Competing interests:** The authors have declared that no competing interests exist.

City and Tokyo are decried in some popular accounts of urban sustainability by virtue of their sheer size, these cities are more energy-efficient than their smaller counterparts [2].

Instead of size, we focus on the internal form and structure of urban settlements, particularly poor street connectivity which we call "street-network sprawl," as a critical determinant of a city's energy and environmental footprint. While street-network sprawl is only one of several dimensions of urban sprawl [3], it is strongly associated with increased vehicle travel, energy use and $CO_2$ emissions [4, 5]. Walking and public transportation are quicker and more attractive on grid-like, connected street networks, while culs-de-sac and gated communities favor driving. Lower population densities increase per capita emissions too [6, 7], but street-network sprawl has an even stronger impact [8]; density and street-network sprawl capture different dimensions of urban spatial structure. Moreover, population densities wax and wane but once streets are laid down, their routes almost never change, even in the face of natural disasters such as earthquakes or fires [9]. Thus, today's choices on the connectivity of streets may lock in patterns of urban development for a century or more.

In contrast to the physical extent of cities or densities, which can be inferred from remote sensing imagery and census data [10–12], street-network sprawl has defied global analysis and quantification. The vast majority of studies, including our own earlier work [5, 13], focus on a single city or country, usually in North America or Europe. While the properties of the street network figure prominently in global typologies of urban structure (notably, [14]) such efforts have been limited to qualitative assessments and/or a small sample of metropolitan regions (see [15] for a prominent exception). Indeed, the lack of comparative, cross-national analysis has been identified as a major research gap in the literature on urban sprawl and its impacts [16]. The urban studies literature more generally has also called for more cross-national comparisons as a critical step for explaining urban processes [17, 18].

In this paper, we conduct the first-ever high-resolution quantification of street connectivity at the global scale, which is now possible through use of OpenStreetMap (OSM)—a crowd-sourced, openly licensed database popularly known as the "Wikipedia of maps". Our previous work [19] shows that OSM road data are largely complete, giving us high confidence in our findings in most regions of the world, with the notable exceptions of China and some parts of Africa. We develop a measure—the Street-Network Disconnectedness Index (SNDi)—that describes similar qualitative attributes of urban form in countries with diverse income levels, cultures and urban planning traditions. We show in this paper that SNDi captures within-country variation, and demonstrates predictive power with respect to cross-sectional variation in transportation energy outcomes.

## 2 Methods

### 2.1 Conceptual issues

The vast majority of empirical studies of street-network sprawl focus on the United States, and measure connectivity using nodal degree, i.e., the number of road network edges joining each intersection, and/or nodal (intersection) density. Together with two measures derived from nodal degree—the proportions of degree-one (i.e., culs-de-sac) and degree-four intersections—such measures are easy to calculate, and demonstrate a strong association with transportation and environmental outcomes. Networks with higher average nodal degree and nodal densities are associated with more walking and cycling, less vehicle travel and lower transportation emissions [5, 8, 20–24]. Intuitively, connected streets provide a more direct path of travel to public transportation and final destinations, and the ensuant shorter travel distances favor non-motorized modes. Street connectivity has also been linked to greater traffic safety

[25], housing price resilience [26] and network resilience [27]. As a result, many local and other sub-national governments promote or mandate greater street connectivity [13, 28].

However, nodal density and degree are unsatisfactory as universal measures of street-network sprawl. While they capture the contrast between historic, walkable urban centers and suburban subdivisions in the North American planning tradition, the measures may not translate to other social and economic contexts. Most importantly, a grid-like road network, where most edges are degree-four, is not the only possible way to achieve high connectivity. Medieval urban centers in places such as Tokyo, Reykjavik, and Athens have irregularly spaced three-way intersections rather than a gridiron pattern, yet have high walkability, low vehicle travel, and, by some measures, connectivity as high as a rectilinear grid [29]. Moreover, urban development can be internally well-connected, but be automobile-oriented through limited connections to the rest of the city, as in a gated community. Nodal degree and nodal density do not necessarily indicate how streets are configured at the network level, and are sensitive to the scale at which an analysis is conducted [25, 29–31].

## 2.2 Global measurement of street-network sprawl

We therefore calculate a suite of measures of street-network sprawl that are derived from graph theoretic and spatial principles, in order to capture the diverse legacy of design traditions in cities around the world. Using the dataset of $\sim$120 million intersections and $\sim$171 million edges, we calculate:

1. *Nodal degree* for each intersection, and three derived aggregate metrics (mean nodal degree, fraction dead-ends and fraction degree-4+);

2. *Dendricity*, i.e. whether a network is tree-like in that there is only one possible route to reach certain nodes;. Specifically, we calculate the fractions of edges that are classified as a network bridge (i.e., a link between two otherwise-disconnected network components), a dead-end, a self loop, or a part of the cycle basis;

3. *Circuity*, or the ratio of path length to straight-line distance between each node and other nodes in its vicinity;

4. *Sinuosity*, or ratio of length to end-to-end distance, for every edge

While the nodal degree and dendricity measures relate purely to the network configuration, our circuity and other measures capture the geographic layout of roads. Fig 1 demonstrates these concepts. Empirical implementations of the last three groups of measures are rare in the literature (for exceptions, see [4, 32–35]), and often use a manual classification procedure that limits the analysis to a few cities or neighborhoods. The S1 File provides a detailed rationale and definitions.

We use a principal components analysis (PCA) to assign empirical weights to our aggregate measures. We focus on the first principal component as an overall measure of street-network sprawl, and we refer to this measure as the Street-Network Disconnectedness Index (SNDi). Higher values of SNDi correspond to increasing street-network sprawl. While SNDi is closely correlated with nodal degree in the United States ($R = 0.91$), it is less so in other countries, notably those in Western Europe such as Germany ($R = 0.71$). Thus, while SNDi is designed to represent the global, high-resolution variation captured by our full set of connectivity measures, it also relates closely to nodal degree in North America and thus to the literature focused there.

The PCA coefficients for SNDi, shown in Table 1, are all positive and are relatively evenly loaded across our measures of sprawl, indicating that the metrics all contribute to SNDi in the

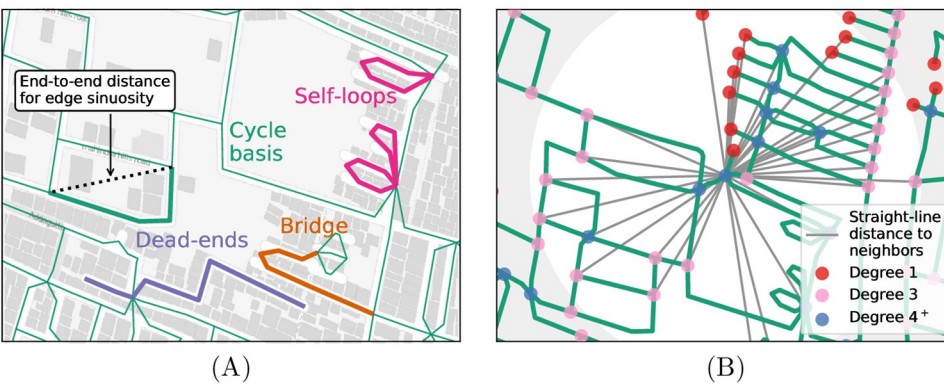

**Fig 1. Schematic of selected connectivity measures.** (a) Edge-related measures include sinuosity, which incorporates the sum of the end-to-end length for each edge, and dendricity metrics, which incorporate a classification of all edges into one of four network-theoretic types. Solid lines indicate streets. (b) Node-related measures are based on the nodal degree, indicated by colored circles, and circuity, which relates to the shortest path lengths and the direct-path distances between one node and all its neighbors within some radius. Green lines indicate streets, while gray lines indicate straight-line distances to nearby nodes within the 500 m radius shown in white.

same direction, and that our varied metrics do capture independent variation in street-network connectivity. In addition, the first component (SNDi) captures the bulk of the variation. The second component, $PCA_2$, in contrast, emphasizes differences between our circuity measures and our other measures of connectivity. Rather than pointing unambiguously in the direction of increasing sprawl, high $PCA_2$ can be interpreted as a place where circuity is much greater than would be expected from nodal degree, dendricity, and sinuosity alone, such as a place where connected neighborhoods are separated by a river or steep topography. Therefore in our primary analysis we focus on SNDi in order to reduce the dimensionality of our analysis. Details, including results and further interpretation for the second component, are in the S1 File.

**Table 1. Principal component loadings.** SNDi corresponds to the $PCA_1$ component.

| | $PCA_1$ | $PCA_2$ | $PCA_3$ |
|---|---|---|---|
| Nodal degree (negative) | .35 | −.12 | −.44 |
| Frc deadends | .35 | −.12 | −.45 |
| Log Circuity (0–0.5km) | .27 | .23 | −.12 |
| Log Circuity (0.5–1km) | .30 | .29 | −.041 |
| Log Circuity (1–1.5km) | .29 | .33 | .030 |
| Log Circuity (1.5–2km) | .25 | .37 | .11 |
| Log Circuity (2–2.5km) | .19 | .38 | .21 |
| Log Circuity (2.5–3km) | .11 | .27 | .32 |
| Frc bridges (length) | .22 | −.34 | .48 |
| Frc non-cycle (length) | .32 | −.31 | .17 |
| Frc non-cycle (N edges) | .38 | −.26 | −.11 |
| Frc bridges (N edges) | .28 | −.30 | .33 |
| Log(sinuosity) | .16 | .033 | .21 |
| Variance explained | 37% | 22% | 9% |
| Eigenvalue | 4.9 | 2.9 | 1.2 |

## 2.3 Data and computational issues

Our representation of the street network is derived from the planetary OSM dataset using a number of preprocessing steps, including splitting roads into edges at intersections, dropping duplicates, merging intersections within 20m of each other, and annealing degree-two intersections into single edges, followed by conversion to a network format conducive to calculating our connectivity metrics. Our earlier work validates the use of OSM to characterize true extant roads in most countries, through quantifying the completeness of the network in every country on Earth [19].

Unless stated, all our results refer to urban areas only, which are the locus of future population and emissions growth. We classify each edge as urban or non-urban based on an imputation involving population density with ∼1 km resolution from LandScan [36], country-specific distributions of density in cities included in the Atlas of Urban Expansion [15], and the nearby fraction of land area developed according to the Global Human Settlements Layer [11].

We use a global grid with ∼1 km resolution to characterize aggregate SNDi at high resolution, and we classify grid cells using $k$-means similarity grouping over eight measures of street-network sprawl.

The S1 File reports results for all areas and discusses our methods in detail, including extensive robustness checks. Complete open-source computer code for quantifying street-network sprawl and reproducing our results are, along with our resulting dataset, linked in the S1 File.

## 3 Results and discussion

### 3.1 Validation of SNDi

We validate how SNDi measures characteristics of the built environment in two ways. First, we construct three abstracted paradigms of urban street networks, in order to demonstrate how certain distinctive patterns of urban form map to SNDi. Second, we undertake a qualitative appraisal to investigate how SNDi relates to recognized characteristics of the built environment.

**3.1.1 Street-network paradigms.** Our three abstracted paradigms are shown in Fig 2 in a similar presentation style to [37]. Our *grid* paradigm is a rectilinear network typical of North and South American city centers, particularly those built prior to the dominance of the private car [38]. Our *medieval* paradigm consists of irregularly spaced streets and non-perpendicular degree-three intersections, and is characteristic of many historic cities in Europe. Our *culs-de-*

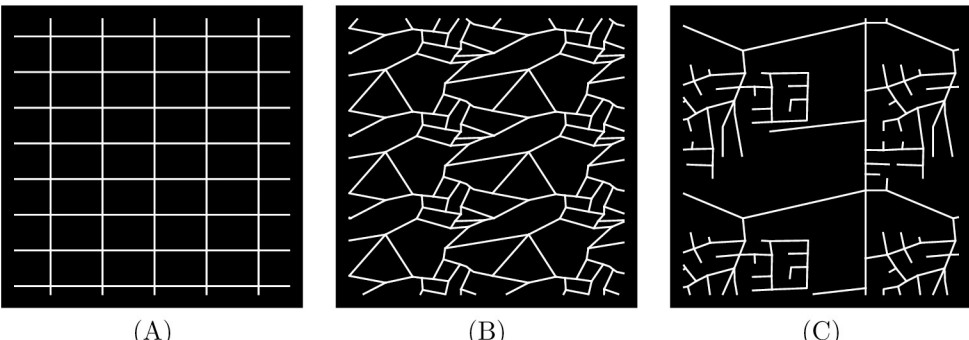

|     |     |     |
| :-: | :-: | :-: |
| (A) | (B) | (C) |

**Fig 2. Three prototypical street-network styles.** (A) The *grid* paradigm has the highest connectivity and an SNDi of -1.01. (B) The *medieval* paradigm is relatively highly connected (SNDi of 0.66) but far from a grid. (C) The *culs-de-sac* paradigm is the most sprawling, with SNDi of 7.9.

*sac* paradigm characterizes the "loops and lollipops" and "lollipops on a stick" street-network designs typical of much mid-to-late 20th Century urban development in North America [13, 38]. Scores for each paradigm on SNDi and its component connectivity measures are provided in the S1 File.

By inspection, the grid and medieval cases have many connections and few dead-ends, and the culs-de-sac case is at the opposite extreme. Such characteristics fail to be captured by nodal degree, on which measure the three paradigms are evenly spaced (4, 3.1, and 2.2 for grid, medieval and culs-de-sac, respectively), or by the fraction of four-way intersections, on which measure the medieval and culs-de-sac cases are almost identical (10% and 9%, respectively, as compared with 100% for grid).

In contrast, our composite measure SNDi captures the high connectivity of *medieval*, while leaving *grid* and *culs-de-sac* near opposite extremes of the global distribution. The low SNDi value for *medieval* (0.66) reflects the zero scores on several measures of dendricity, and circuity scores which are nearly as low as for *grid*. That is, our composite measure captures the absence of dendritic branches and dead-ends in the irregular *medieval* network, which is composed largely of three-way intersections, and the property that its network distances tend not to diverge radically from straight-line distances. By contrast, the *culs-de-sac* case, with SNDi of 7.9, suffers from "one-way-out" regions, dead-ends, sinuous edges, and highly circuitous travel paths. These features are all well captured by our suite of street-network sprawl metrics and, in turn, by SNDi.

**3.1.2 Qualitative appraisal.** We use satellite and street-level imagery to examine how our measure of street-network sprawl relates to network connectivity along with other attributes emphasized in the walkability literature that SNDi does not address directly, such as the mix of residential and commercial uses and the presence of sidewalks or footpaths [39]. We explore automatically-chosen examples at the 5th and 95th global percentiles of SNDi in the world's ten largest countries. Fig 3 shows the images from two contrasting countries, the United States and Indonesia; the full set, including examples at the median as well as the 5th and 95th percentiles, is provided in Table D in S1 File.

The examples in the upper row of Fig 3 show high-connectivity neighborhoods in an inner suburb of Seattle, USA, and in the coastal town of Kolaka—a small town in South-East Sulawesi ∼1600 km northeast of Jakarta. In Seattle, the streets are characterized by a range of residential and commercial uses, including single-family homes, two- to six-story apartment buildings, restaurants, grocery stores, religious institutions, schools and other services; the grocery store pictured in the linked Google Streetview image, with apartments above and a sidewalk cafe, is typical of the neighborhood. Vehicle parking is often in the rear, allowing for a continuous sidewalk uninterrupted by driveway access. WalkScore, a proprietary walkability rating system based on distances to common destinations, gives the neighborhood a "walker's paradise" score of 98/100. In Kolaka, the neighborhood is mixed-use, with numerous informal and formal businesses occupying the ground floor of the 1–2 storey buildings, and some stand-alone commercial buildings interspersed with residential and civic uses. Unusually for Indonesia, sidewalks are present at least intermittently on some major streets. In both cities, the grid is not regular but most intersections are degree-four, and there are few culs-de-sac.

The lower row of Fig 3 shows low-connectivity neighborhoods in Orange County, California and one of Jakarta's southern suburbs. The Orange County example is characterized by intermittent sidewalks and a streetscape dominated by garage doors which serve as the only street-facing entrances to houses. The retail complex in the lower right corner of the satellite image has no direct pedestrian connection to the adjacent residential neighborhood, and the gated community (with guard) to its north further curtails pedestrian connections. The WalkScore is 21/100 ("car-dependent"), and even this rating reflects an algorithm that

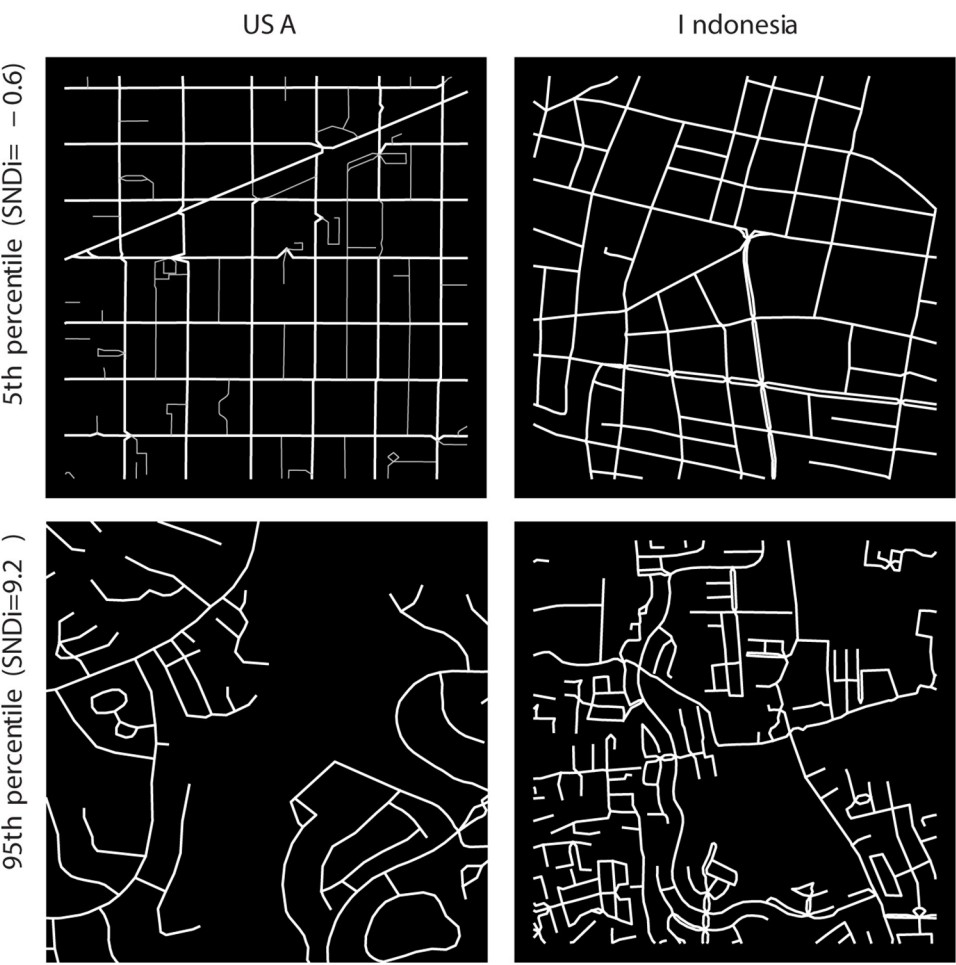

**Fig 3. Examples of high and low street-network sprawl.** Street network diagrams from sample neighborhoods in two countries, restricted to the 30-arc second grid cell. A version with satellite and Google Streetview imagery integrated is available online. The upper and lower rows show examples at the 5th and 95th global percentiles of SNDi respectively. The S1 File contains a large set of examples.

inflates the neighborhood's connectedness through using straight-line rather than network distances. The suburban Jakarta example also features a series of guard-gated residential communities of different income levels, which are largely cut off from each other due to the combination of street-network design and physical constraints such as watercourses. The more recent development, catering to middle-class households, is typified by detached houses with parking between the front of the house and the street. The surrounding lower-income, mixed-use community has a range of street stalls and ground-floor commercial uses, but still features numerous dead ends. In neither Orange County nor suburban Jakarta are sidewalks present.

Our qualitative assessment here and in the S1 File indicates that SNDi correlates with different aspects of walkability and street connectivity in a range of economic and geographic contexts. Numerous related aspects of street character and urban design highlighted by this assessment, such as sidewalk widths and street-facing doors and windows, lend themselves to more focused analysis in future work, but themselves are likely to be constrained by the overall connectivity of the network.

## 3.2 Empirical street-network types

SNDi provides a single-dimensional measure of street-network sprawl, and thus does not differentiate between the different avenues to a given level of connectivity. For example, as previously stated, good connectivity does not necessarily dictate a gridiron pattern. We therefore develop a multidimensional classification that identifies empirical street-network types (or "types" for short) that emerge from the data. We use $k$-means cluster analysis to partition the $\sim 4.4$ million 30-arc second ($\sim 1 km^2$) urban grid cells into eight types (for details, see S1 File).

The types, shown in Fig 4, are ordered by SNDi but differ qualitatively according to particular metrics. Types A and C are regular and irregular grids respectively, while Type B is differentiated from grid cells of similar SNDi by a high proportion of degree-3 nodes. Type E is characterized by circuitous road networks, and many grid cells in this type are gated communities, such as the example shown from the Philippines. Type F is characterized by a high fraction of deadends, while Type G, illustrated by a suburb of Seoul, has a dendritic network and has the highest fraction of network bridges. Type H is the least connected on almost every dimension, and has the highest proportion of deadends and the most circuitous road networks. Additional examples of each type and the centroid values defining them are given in the S1 File.

Types B and C are prevalent worldwide and have almost identical mean values of SNDi, indicating the alternative designs that can provide reasonable connectivity. Similarly, types E and F have similar mean values of SNDi, with E having greater circuity, but not necessarily the large fraction of deadends that typifies F. Type G has a very broad SNDi distribution, indicating that dendritic street-network sprawl manifests itself in alternative ways. The majority of

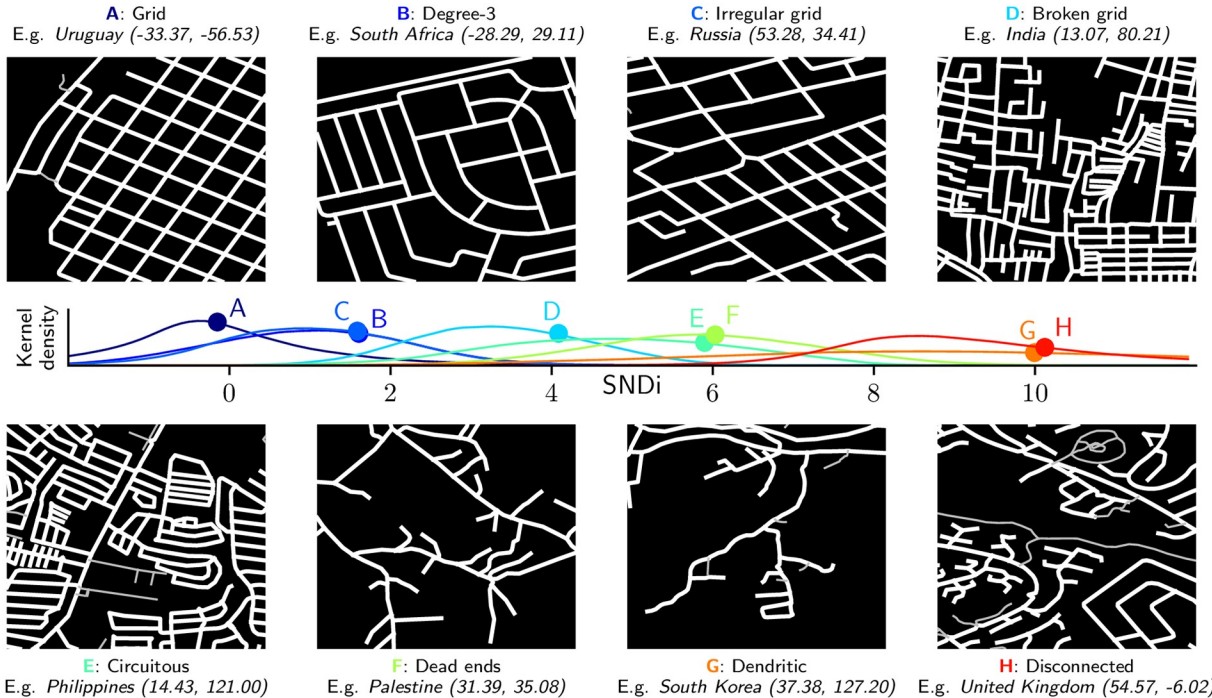

**Fig 4. Empirical street-network types.** The eight types are identified through $k$-means cluster analysis of 30-arc second grid cells, and ordered by increasing SNDi. For each type, streets in an example grid cell near the type's centroid are shown, along with a plot of the type-level distributions (colored lines) and means (colored circles) of SNDi across grid cells. The caption indicates the (latitude, longitude) of each example grid cell as well as the country where it is located. Service roads and paths that are closed to cars (most visible in Type H) are depicted with thinner grey lines.

types, however, have fairly symmetrical loadings—for example, a type with a circuitous street network also tends to have more dead-ends—providing evidence that our street-network sprawl metrics covary over a qualitative variety of developments.

### 3.3 Street-network sprawl and automobility

The theory and empirical evidence discussed in Section 1 suggest that both SNDi and the more connected empirical types should be associated with higher levels of vehicle ownership and travel, and with less walking. The association is likely to be strongest in high-income countries where vehicle ownership is saturating, and weakest in low-income countries where most households cannot afford a private car and walking may be a necessity rather than a choice. In lower-income places, we would predict the influence of already-cemented street-network connectivity on automobility to increase as incomes rise over time, as suggested by [40, 41].

We therefore examine the associations between household car ownership and SNDi in several high-income countries spanning different continents and with data availability at subnational geographies: the United States, Japan, New Zealand, and France (see S1 File for data sources). Three primary conclusions can be drawn from the regressions shown in Table 2. First, SNDi is strongly associated with car ownership; one standard deviation increase in SNDi increases vehicle ownership by 0.35 standard deviations or 0.16 cars household$^{-1}$ in the univariate case (model 1), and by 0.25 standard deviations or 0.12 cars even when controlling for density (model 2). Second, SNDi is capturing a dimension of urban form separate from that of density; both variables are highly predictive when included in the regression together. Third, SNDi has similar predictive power to a traditional measure of street-network connectivity, nodal degree (model 3), which may reflect the high correlation between nodal degree and SNDi in the United States in particular. Similar results (models 4-6) are obtained for the associations between the fraction of workers commuting by walking (the data for which are limited to the United States and New Zealand). Further robustness tests are discussed in the S1 File.

Our empirical types (Section 3.2) also discriminate strongly between high- and low- automobility regions. In the United States, where the automobility data are available at high geographic resolution, low car ownership and high proportions of walking commutes are disproportionately found in our gridded types A and C, while high car ownership and little walking tend to be concentrated in the least-connected types E, F, G and H (Fig 5). Related plots (Fig K in S1 File) show the kernel density and cumulative distributions of car ownership and walking by empirical type, and yield a similar conclusion. Our results do not necessarily

**Table 2. Regressions of vehicle ownership and walking.** Coefficients shown are standardized (beta) coefficients, with robust standard errors in parentheses. All *p*-values are less than 0.1%. Constant terms are not shown. All coefficients are statistically significant at conventional levels. All models include fixed effects for country (United States, France, New Zealand and Japan). For data sources and units of analysis, see the S1 File.

| | Cars per household | | | Fraction walking | | |
|---|---|---|---|---|---|---|
| | (1) | (2) | (3) | (4) | (5) | (6) |
| SNDi | .35 | .25 | .14 | −.25 | −.23 | −.11 |
| | (.002) | (.002) | (.003) | (.002) | (.002) | (.004) |
| Nodal degree ×(−1) | | | .14 | | | −.14 |
| | | | (.004) | | | (.006) |
| log(density) | | −.42 | −.39 | | .091 | .066 |
| | | (.002) | (.002) | | (.003) | (.003) |
| country fixed effects | ✓ | ✓ | ✓ | ✓ | ✓ | ✓ |
| $R^2_{adjusted}$ | .276 | .438 | .442 | .063 | .071 | .075 |
| N | 231800 | 231800 | 231800 | 214800 | 214800 | 214800 |

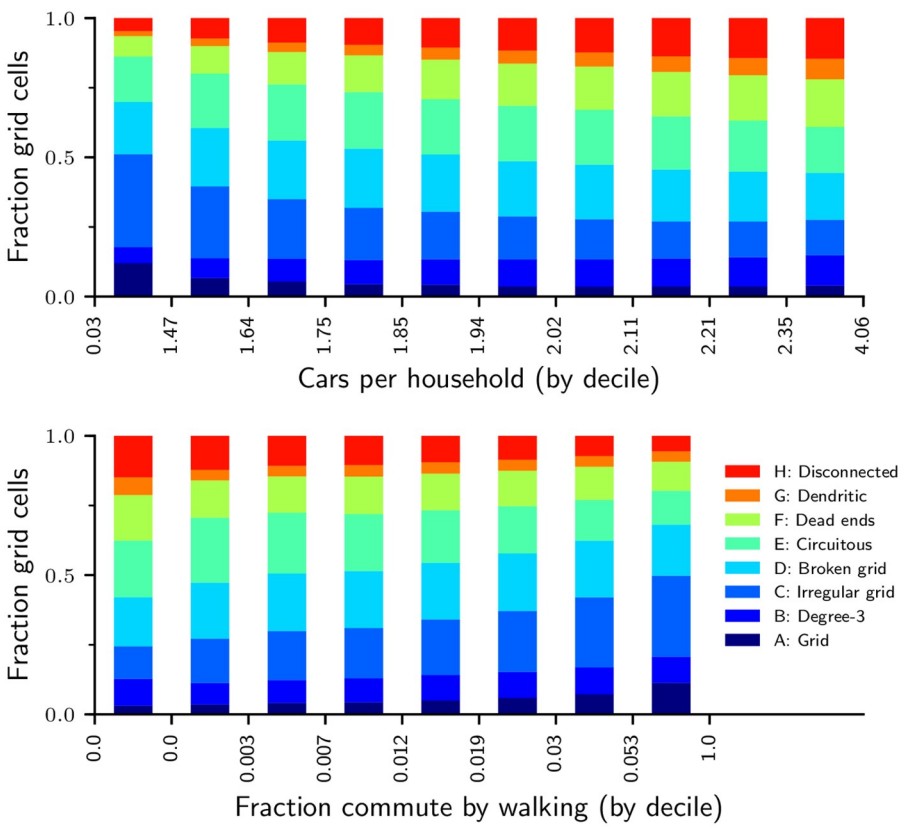

**Fig 5. Distribution of empirical types by vehicle ownership and walking, United States.** Vehicle ownership and commute mode share are aggregated to grid cells, based on an area-weighted average of census block groups that intersect a given grid cell. The lower two deciles are not defined for fraction commute by walking; because ∼30% of the grid cells have a zero walk share, the 1st and 2nd deciles are combined with the 3rd decile.

identify a causal connection, but they do reinforce the validity of our approach to measuring urban form, in that both SNDi and the types are associated with household transportation decision-making and ultimately with the determinants of transportation energy use and emissions.

## 3.4 Patterns of street-network sprawl

We now turn to analyzing cross-sectional variation in SNDi around the world (Fig 6, Tables 3, and 4). The most-connected streets are found in South America, where the grid-like patterns can be at least partly attributed to the Spanish colonial legacy [42]. Uruguay, Argentina and Paraguay lie towards the bottom of a country-level ranking of SNDi, with average nodal degree of >3.2. Japan, South Korea, much of Europe and North Africa also stand out for their low levels of street-network sprawl, although especially in the Japanese case, the English-language literature offers fewer clues about the historical origins of these patterns.

Higher levels of SNDi are found in the United States, confirming numerous popular and academic accounts [43], but more surprisingly in the United Kingdom, Ireland and Norway as well, where disconnected street patterns have attracted little or no attention. A cluster of countries with high street-network sprawl is found in southeast and south Asia, where Sri Lanka, the Philippines, Bangladesh, Indonesia, Thailand, Malaysia and Vietnam all lie within the top quartile of country-level SNDi. In cities such as Manila and Jakarta, one potential explanation

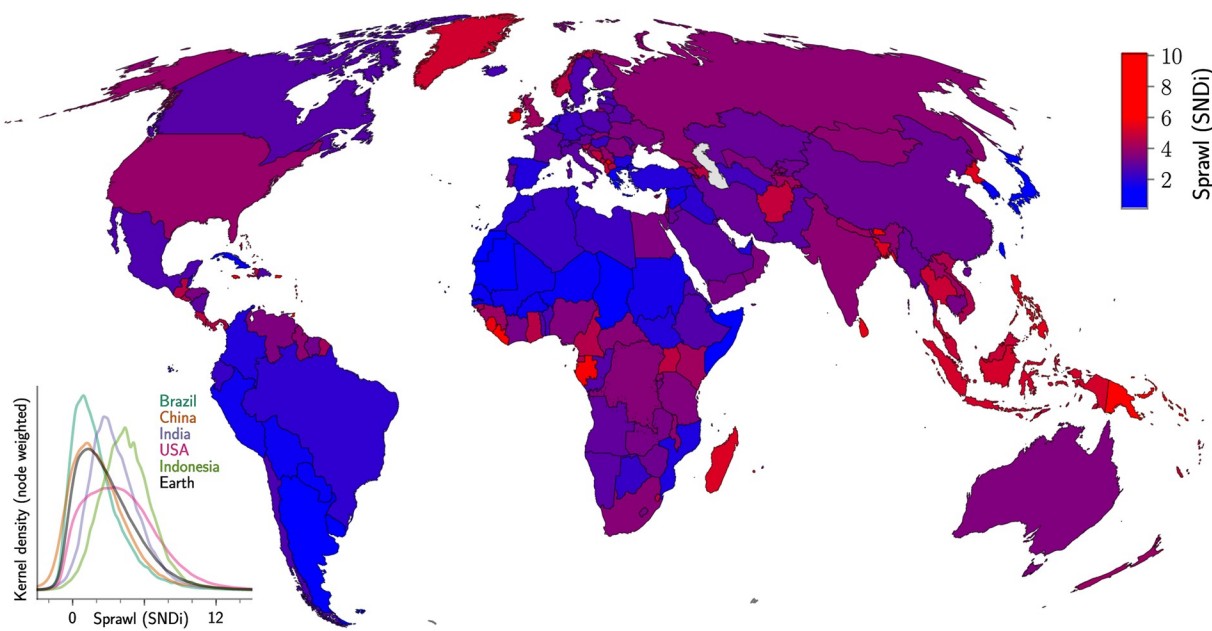

**Fig 6. Urban street-network sprawl (SNDi) by country, 2018.** The inset charts the within-country distribution of SNDi for the largest countries by population. The distributions are grid cells weighted by the number of nodes in each cell; see Fig F in S1 File for unweighted distributions and distributions for additional countries.

**Table 3. Countries listed by street-network connectivity.** The top ten and bottom ten countries with ≥10M population and ≥100k street nodes, ordered by SNDi. Countries with populations >20M are listed in bold. A fuller list is in Table F in S1 File.

|  | Sprawl (SNDi) | Nodal degree | Frc non-cycle (length) | Log Circuity (0.5–1km) | N(nodes) |
|---|---|---|---|---|---|
| **Sri Lanka** | 5.5 | 2.5 | 0.2 | 0.2 | 150k |
| **Philippines** | 5.4 | 2.8 | 0.3 | 0.2 | 500k |
| **Bangladesh** | 5.3 | 2.6 | 0.3 | 0.2 | 160k |
| **Indonesia** | 5.1 | 2.7 | 0.3 | 0.2 | 1.6M |
| **Thailand** | 4.9 | 2.7 | 0.2 | 0.2 | 920k |
| **Malaysia** | 4.9 | 2.8 | 0.2 | 0.2 | 420k |
| **Uganda** | 4.7 | 2.6 | 0.2 | 0.2 | 120k |
| **Ghana** | 4.6 | 2.7 | 0.2 | 0.2 | 130k |
| Guatemala | 4.6 | 2.9 | 0.3 | 0.2 | 110k |
| **Cameroon** | 4.5 | 2.7 | 0.2 | 0.2 | 150k |
| ⋮ | ⋮ | ⋮ | ⋮ | ⋮ | ⋮ |
| **Syrian Arab Republic** | 1.7 | 3.0 | 0.1 | 0.1 | 190k |
| **Sudan** | 1.6 | 3.3 | 0.3 | 0.1 | 320k |
| **Peru** | 1.5 | 3.2 | 0.1 | 0.1 | 280k |
| Greece | 1.5 | 3.1 | 0.1 | 0.1 | 380k |
| Bolivia | 1.5 | 3.2 | 0.1 | 0.1 | 170k |
| Senegal | 1.5 | 3.2 | 0.1 | 0.1 | 150k |
| Cuba | 1.4 | 3.1 | 0.1 | 0.1 | 100k |
| **Korea, Rep.** | 1.4 | 3.2 | 0.1 | 0.1 | 370k |
| **Japan** | 1.3 | 3.1 | 0.1 | 0.1 | 4M |
| **Argentina** | 0.6 | 3.4 | 0.0 | 0.1 | 820k |

**Table 4. Cities listed by street-network connectivity.** The top ten and bottom ten cities with ≥3,000 street nodes, ordered by SNDi. A fuller list is in Table G in S1 File.

| | | Sprawl (SNDi) | Nodal degree | Frc non-cycle (length) | Log Circuity (0.5–1km) | N(nodes) |
|---|---|---|---|---|---|---|
| Bangkok | THAILAND | 7.3 | 2.5 | 0.3 | 0.3 | 150k |
| Cebu City | PHILIPPINES | 6.6 | 2.6 | 0.3 | 0.3 | 12k |
| Raleigh | UNITED STATES | 5.6 | 2.6 | 0.2 | 0.2 | 51k |
| Caracas | VENEZUELA | 5.3 | 2.8 | 0.2 | 0.3 | 12k |
| Palembang | INDONESIA | 5.2 | 2.6 | 0.3 | 0.2 | 39k |
| Kampala | UGANDA | 5.1 | 2.6 | 0.2 | 0.2 | 15k |
| San Salvador | EL SALVADOR | 5.0 | 2.7 | 0.2 | 0.2 | 17k |
| Medan | INDONESIA | 4.9 | 2.6 | 0.3 | 0.2 | 64k |
| Pune | INDIA | 4.9 | 2.7 | 0.2 | 0.2 | 31k |
| Rawang | MALAYSIA | 4.7 | 2.9 | 0.1 | 0.3 | 5.1k |
| ⋮ | ⋮ | ⋮ | ⋮ | ⋮ | ⋮ | |
| Vienna | AUSTRIA | 1.0 | 3.3 | 0.1 | 0.1 | 23k |
| Berlin | GERMANY | 0.9 | 3.3 | 0.0 | 0.1 | 40k |
| Hong Kong | CHINA | 0.9 | 3.4 | 0.0 | 0.1 | 6.1k |
| Gwangju | SOUTH KOREA | 0.9 | 3.3 | 0.1 | 0.1 | 10k |
| Cheonan | SOUTH KOREA | 0.8 | 3.3 | 0.1 | 0.1 | 5.3k |
| Kairouan | TUNISIA | 0.6 | 3.2 | 0.0 | 0.1 | 3.2k |
| Thessaloniki | GREECE | 0.5 | 3.3 | 0.1 | 0.1 | 14k |
| Khartoum | SUDAN | 0.5 | 3.3 | 0.0 | 0.1 | 110k |
| Buenos Aires | ARGENTINA | 0.3 | 3.5 | 0.0 | 0.1 | 160k |
| Valledupar | COLOMBIA | 0.2 | 3.5 | 0.0 | 0.1 | 4.8k |

may lie in the rise of a new middle class that has retreated to gated communities in response to fears of crime, inadequate public services, and weak land-use regulation. Since the 1950s, urban expansion in these cities has been characterized by walled subdivisions with privately provided services such as water, garbage removal, security and recreation, and little to no access by public transit or walking; poor street connectivity can be seen as a way to generate social and physical exclusivity [44, 45]. In Kuala Lumpur and Bangkok, while gated communities are less common, urban expansion has been led by private firms, generating developer-planned townships and subdivisions with few connections to their immediate neighbours [46].

There is also considerable variation within countries. The inset to Fig 6 shows the distribution of SNDi for the largest five countries by population. The widest distribution is found in the United States; while the country is home to some of the least-connected streets on the planet, it also has examples at the opposite end of the SNDi spectrum. Otherwise, the distribution largely reflects the ordering evident from the map, with Brazil and China having the lowest levels of street-network sprawl among the largest countries, and Indonesia the highest. Distributions for additional countries, along with further sub-national results, are provided in the S1 File.

Patterns of street-network sprawl can also be understood through the distribution of the empirical types (defined in Section 3.2), as shown in Fig 7 for the largest ten countries and cities. In China, Brazil and Japan, and most markedly in Tokyo and Buenos Aires, the grid, degree-3 and interrupted grid types (A, B and C) account for more than half of the urbanized area. The most-sprawling dendritic and disconnected types (G and H) are almost completely absent in Japan as a whole and in Tokyo and Buenos Aires specifically. Overall, however, a notable finding from Fig 7 is how each of the eight types is found in virtually every large

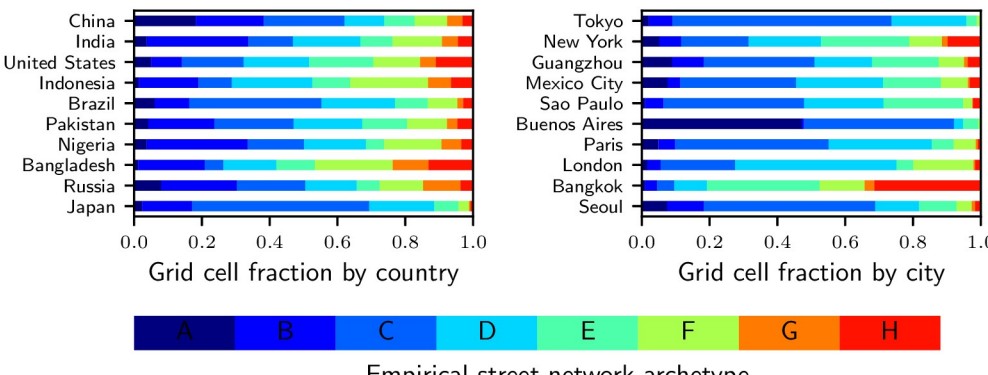

**Fig 7. Distribution of empirical types.** Distributions are shown for the largest ten countries (by population) and cities (by length of edges, restricted to one per country). Other countries and cities are shown in Figs H, I and J in S1 File. City extents are defined using [15].

country and city. While countries with high SNDi have a greater prevalence of the sprawling types, grids and other more-connected types are not completely absent.

## 4 Conclusion

A large literature in urban planning, economics and related disciplines highlights the critical importance of urban form in shaping individual decisions on vehicle ownership, use, and energy consumption. The lack of globally consistent measures of urban form, however, has hampered comparative studies, and limited the ability of researchers and policy makers to understand the structural determinants of transportation energy consumption and associated emissions, as well as the other mechanisms through which urban spatial structure affects the economic and social organization of cities.

In constructing globally consistent measures of urban form, our focus on street connections is based on the well-established principle that disconnected streets promote travel by private car and discourage walking. We propose that globally comparable and meaningful metrics of street-network sprawl should (1) relate to similar qualitative attributes of the built environment across countries; (2) vary meaningfully within countries as well as across countries; and (3) have predictive power with respect to cross-sectional variation in transportation and energy outcomes. Based on these criteria, our study validates a univariate measure, the SNDi, and a multidimensional classification, which incorporate graph-theoretic and geographic concepts, and can be computed for every mapped intersection and street segment on the planet. While some of the eight empirical types in our classification are more common in particular countries, each type is found across a diversity of geographic regions (Section 3.3), illustrating the regularities in patterns of street-network sprawl around the world.

Theoretical and empirical literature [8] suggests that street-network sprawl should have substantial long-run impacts on energy use for urban transportation. While our analysis here leaves much room for future work, we show that SNDi and our empirical types are strongly associated with car ownership and walking within select high-income countries with different energy prices, urban planning traditions, and political and economic histories. Moreover, SNDi outperforms traditional measures of connectivity (e.g. nodal degree), which cannot be expected to, and do not appear to, travel well outside the United States where such measures were originally applied. There are multiple ways in which planners and developers can promote connected streets, not just the gridiron patterns commonly found in North and South America.

We have argued [5, 13] in the context of the United States that street-network sprawl is a path-dependent process. Because streets are one of the most permanently defining features of cities, today's decisions on street-network connectivity are likely to have compounded influence on land use and transportation outcomes over time, and policy makers' responsibility encompasses the long-term effects of contemporary choices by developers and planners. We speculate that, globally, compounding will be even stronger in places such as Jakarta (Fig 3), where income may currently be the main constraint on car ownership but street-network sprawl will assume an increasingly important role as affluence grows and car ownership and orientation become a choice to more residents. Addressing density and public transit access at the time of new construction is important, but these are relatively plastic features of urban form, while the road network is likely to act as a more permanent constraint.

This paper shows that there is wide variation in street-network sprawl among the most populous and fastest growing countries in the global South, where the rate of urban growth means that today's decisions will have substantial implications for global energy use and climate change. The Philippines and Indonesia are among the most sprawling by our analysis, while some of the least sprawling countries are in Latin America and the Middle East. Our analysis provides a framework and data for future research to help to understand the reasons for such divergence, and how urban growth policies can move cities onto a less sprawling path.

## Supporting information

**S1 File. Supporting information, including additional documentation, figures and tables.**
(PDF)

**S2 File. Code for replication of the study's results.**
(BZ2)

## Acknowledgments

We are grateful for excellent research assistance from Tássia Araújo, Gal Kramer, and Sabina Sloman, and to Elliott Campbell, John Armstrong, Nazanin Rezaei, Paulo Quadri, Rachel Voss, Ruihua Wang and Kai Zhu for helpful comments on earlier drafts. Our work was supported by Social Science and Humanities Research Council of Canada grant 435-2016-0531, the Hellman Fellows Program, and a UC Santa Cruz Faculty Research Grant. Most importantly, we thank the ∼5 million contributors to the OpenStreetMap dataset. This article uses the LandScan 2012 global population data set from Oak Ridge National Laboratory.

## Author Contributions

**Conceptualization:** Christopher Barrington-Leigh, Adam Millard-Ball.

**Funding acquisition:** Christopher Barrington-Leigh.

**Investigation:** Christopher Barrington-Leigh, Adam Millard-Ball.

**Methodology:** Christopher Barrington-Leigh, Adam Millard-Ball.

**Software:** Christopher Barrington-Leigh, Adam Millard-Ball.

**Validation:** Christopher Barrington-Leigh, Adam Millard-Ball.

**Writing – original draft:** Christopher Barrington-Leigh, Adam Millard-Ball.

**Writing – review & editing:** Christopher Barrington-Leigh, Adam Millard-Ball.

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
