## [Decision Letter · Decision Letter 0]

25 Jul 2019

PONE-D-19-15599

A global assessment of street-network sprawl

PLOS ONE

Dear Dr. Millard-Ball,

Thank you for submitting your manuscript to PLOS ONE. After careful consideration, we feel that it has merit but does not fully meet PLOS ONE’s publication criteria as it currently stands. Therefore, we invite you to submit a revised version of the manuscript that addresses the points raised during the review process.

We would appreciate receiving your revised manuscript by Sep 08 2019 11:59PM. To enhance the reproducibility of your results, we recommend that if applicable you deposit your laboratory protocols in protocols.io, where a protocol can be assigned its own identifier (DOI) such that it can be cited independently in the future. For instructions see: http://journals.plos.org/plosone/s/submission-guidelines#loc-laboratory-protocols

We look forward to receiving your revised manuscript.

Kind regards,

Hironori Kato, Dr. Eng.

Academic Editor

PLOS ONE

**Journal Requirements:**

2. Our internal editors have looked over your manuscript and determined that it is within the scope of our Urban Ecosystems Call for Papers. This collection of papers is headed by a team of Guest Editors for PLOS ONE. The Collection will encompass a diverse range of research articles on urban environments.  Additional information can be found on our announcement page: https://collections.plos.org/s/urban-ecosystems.

If you would like your manuscript to be considered for this collection, please let us know in your cover letter and we will ensure that your paper is treated as if you were responding to this call. If you would prefer to remove your manuscript from collection consideration, please specify this in the cover letter.

3.  We note that  Figure(s) in your submission contain [map/satellite] images which may be copyrighted. All PLOS content is published under the Creative Commons Attribution License (CC BY 4.0), which means that the manuscript, images, and Supporting Information files will be freely available online, and any third party is permitted to access, download, copy, distribute, and use these materials in any way, even commercially, with proper attribution. For these reasons, we cannot publish previously copyrighted maps or satellite images created using proprietary data, such as Google software (Google Maps, Street View, and Earth). For more information, see our copyright guidelines: http://journals.plos.org/plosone/s/licenses-and-copyright.

a) You may seek permission from the original copyright holder of Figure(s) [#] to publish the content specifically under the CC BY 4.0 license.  

**Additional Editor Comments (if provided):**

Two external reviewers suggested “major revision” and “accept,” respectively. Both reviewers showed positive opinions regarding the study’s academic contribution. I also agree to them. Meanwhile, Reviewer 1 raises some important questions/comments and I feel that they could be helpful to upgrade your manuscript significantly. Then, I am willing to make a decision of major revision. I expect you to incorporate them smartly into your manuscript through revisions.

**Comments to the Author**

1. Is the manuscript technically sound, and do the data support the conclusions?

Reviewer #1: Partly

Reviewer #2: Yes

2. Has the statistical analysis been performed appropriately and rigorously? 

Reviewer #1: Yes

Reviewer #2: Yes

3. Have the authors made all data underlying the findings in their manuscript fully available?

Reviewer #1: Yes

Reviewer #2: Yes

4. Is the manuscript presented in an intelligible fashion and written in standard English?

Reviewer #1: Yes

Reviewer #2: Yes

5. Review Comments to the Author

Reviewer #1: This is an interesting paper and does a nice job building on the authors’ past work to explore street networks. It requires some revision to be suitable for publication.

In section 2.2 you refer to “sinuosity” as an indicator, but in table 1 you refer to “curviness.” Are these the same indicator? Clarification needed.

I’m curious about the motivation to use only the first PCA component as the index, when it only explains about one-third of the variance. I can understand the desire for a single-dimensional indicator, rather than two or three, but what is being lost or ignored if it only explains 36% of the variance? How effective of an indicator is it in the end? I would like to see this more clearly unpacked and motivated as a standalone indicator.

The paper doesn’t have a traditional literature review section per se, but references to the literature appear in some of the background developed in sections 1 and 2. It would be useful to see more clearly how this work is positioned in the research stream, including your own work. For example, this appears to build on the past work of your own that you cite here in PNAS, ERL, and PLOS. Looking through those papers, I see some methodological and conceptual commonalities with this work, but also a nice trajectory leading to this project. It might be useful to explain how this all fits together.

This paper also has similarities to a couple of other street network analysis studies I saw recently, which should appear in your intro/methods sections, especially considering your call for larger sample sizes or global comparative work, which overlaps with theirs. The first did a descriptive analysis of American street networks and included some visuals similar to your figure 2: https://doi.org/10.1177/2399808318784595 The second was an analysis across countries/regions and included a clustering component to identify network design paradigms, similar to yours: https://doi.org/10.31235/osf.io/qj3p5 Your study does a nice job taking things a step further (with the auto-ownership analysis) than the descriptive/clustering analyses of those two papers, creating a nice next step in the overall literature trajectory.

On page 5 you mention empirical implementations of circuity measures (etc) being rare in the research literature and provide a couple exceptions. In addition, see also the two studies linked above (they also appear to operationalize a circuity indicator), as well as these other studies: https://doi.org/10.1068%2Fb130131p and https://doi.org/10.1016/S0965-8564(01)00044-1

I like your multiple-scales circuity measure. Is that a new contribution in this paper? If so, it should be mentioned as such because it seems like a useful way of considering it.

Page 5: by “network bridge” do you mean a bridge in the graph theory sense? It might be worth making that explicitly, as “bridge” terminology can get overloaded when considering road networks (which have overpasses and underpasses).

Page 7: you mention merging compound intersections and other preprocessing steps that are not explained in the paper text, but presumably would substantially impact results. You don’t need to reproduce everything from the SI here, but it would be helpful if you could explain how/why you did the key preprocessing steps for comprehensibility’s sake. For example, how did you define and merge compound intersections? What parameters did you use? What do you mean by “annealing joints”? I think a short summary would do.

Page 7: if your results examine urban areas only, what is the purpose of conducting all the data collection and analysis for the entire planet?

Section 3.1.2: the qualitative assessment suggests that there’s a lot more to street character beyond the network indicators you’ve quantified here. How does that affect the interpretation of your findings? What is being lost when we quantify “the urban” that simply does not always lend itself to quantitative measurement?

It’s not clear in the SI why you chose to parameterize the cluster analysis with k=8. How was this selected and why? Is this preferable to something like hierarchical clustering that allows flexible cluster sizes and a more interpretative cut-off process for the cluster number? This needs to be more clearly motivated.

Table 2: are your region fixed effects referring to the country? Or to subregion entities within the countries?

Reviewer #2: Actually this paper shows original and interesting classification such as Figure 4, linking between real road type and road data from Open Street Map. This classification has very important meanings that can be easily evaluated by global data such as OSM. The statistical result looks feasible and realistic. So I like this paper and can accept without revision.

6. PLOS authors have the option to publish the peer review history of their article (what does this mean?). If published, this will include your full peer review and any attached files.

Reviewer #1: No

Reviewer #2: No

---

## [Author Response · Author response to Decision Letter 0]

23 Aug 2019

Please see the detailed response uploaded as a PDF.

---

## [Decision Letter · Decision Letter 1]

11 Sep 2019

PONE-D-19-15599R1

A global assessment of street-network sprawl

PLOS ONE

Dear Dr. Millard-Ball,

Thank you for submitting your manuscript to PLOS ONE. After careful consideration, we feel that it has merit but does not fully meet PLOS ONE’s publication criteria as it currently stands. Therefore, we invite you to submit a revised version of the manuscript that addresses the points raised during the review process.

We would appreciate receiving your revised manuscript by Oct 26 2019 11:59PM. To enhance the reproducibility of your results, we recommend that if applicable you deposit your laboratory protocols in protocols.io, where a protocol can be assigned its own identifier (DOI) such that it can be cited independently in the future. For instructions see: http://journals.plos.org/plosone/s/submission-guidelines#loc-laboratory-protocols

We look forward to receiving your revised manuscript.

Kind regards,

Hironori Kato, Dr. Eng.

Academic Editor

PLOS ONE

Additional Editor Comments (if provided):

Reviewer 1 suggested “minor revision” with short comments. I also agree to the comments. Please make revisions following the comments.

Reviewers' comments:

Reviewer's Responses to Questions

**Comments to the Author**

1. If the authors have adequately addressed your comments raised in a previous round of review and you feel that this manuscript is now acceptable for publication, you may indicate that here to bypass the “Comments to the Author” section, enter your conflict of interest statement in the “Confidential to Editor” section, and submit your "Accept" recommendation.

Reviewer #1: (No Response)

2. Is the manuscript technically sound, and do the data support the conclusions?

Reviewer #1: Yes

3. Has the statistical analysis been performed appropriately and rigorously? 

Reviewer #1: Yes

4. Have the authors made all data underlying the findings in their manuscript fully available?

Reviewer #1: Yes

5. Is the manuscript presented in an intelligible fashion and written in standard English?

Reviewer #1: Yes

6. Review Comments to the Author

Reviewer #1: The revised manuscript looks good in general. A couple quick comments.

The author’s response regarding the first PCA component makes sense. I’m still a little concerned about its low percent of variance explained (in terms of real-world interpretation), but at the same time I believe that your next text in the revision sufficiently caveats and explains your finding.

Regarding your comments on the lit review: it seems that this is the standard PLOS format and is therefore fine. However, I do believe that leaving out the reference to Ballou et al 2002 remains an oversight, especially considering your claim about the rarity of empirical implementations of circuity. I encourage you to reconsider.

7. PLOS authors have the option to publish the peer review history of their article (what does this mean?). If published, this will include your full peer review and any attached files.

Reviewer #1: No

---

## [Author Response · Author response to Decision Letter 1]

11 Sep 2019

Dear Dr. Kato,

Thank you for the opportunity to submit a revised version of our manuscript, “A global assessment of street-network sprawl,” to PLOS ONE. We thank Reviewer #1 for their comments, and respond as follows:

Reviewer comment: The revised manuscript looks good in general.

Response: Thank you.

Reviewer comment: The author’s response regarding the first PCA component makes sense. I’m still a little concerned about its low percent of variance explained (in terms of real-world interpretation), but at the same time I believe that your next text in the revision sufficiently caveats and explains your finding.

Response: Thank you. We are glad that the additional text provides sufficient justification and caveats.

Reviewer comment: Regarding your comments on the lit review: it seems that this is the standard PLOS format and is therefore fine. However, I do believe that leaving out the reference to Ballou et al 2002 remains an oversight, especially considering your claim about the rarity of empirical implementations of circuity. I encourage you to reconsider.

Response: We have added the Ballou et al 2002 reference to the main text.

Chris Barrington-Leigh and Adam Millard-Ball

---

## [Editor Report · Decision Letter 2]

13 Sep 2019

A global assessment of street-network sprawl

PONE-D-19-15599R2

Dear Dr. Millard-Ball,

We are pleased to inform you that your manuscript has been judged scientifically suitable for publication and will be formally accepted for publication once it complies with all outstanding technical requirements.

With kind regards,

Hironori Kato, Dr. Eng.

Academic Editor

PLOS ONE

Additional Editor Comments (optional):

Thank you for your revisions.
---

## [Editor Report · Acceptance letter]

13 Nov 2019

PONE-D-19-15599R2 

A global assessment of street-network sprawl 

Dear Dr. Millard-Ball:

I am pleased to inform you that your manuscript has been deemed suitable for publication in PLOS ONE. Congratulations! Your manuscript is now with our production department. 

With kind regards,

on behalf of

Dr. Hironori Kato 

Academic Editor

PLOS ONE